# Analyzing the Fitting of Novel Preformed Osteosynthesis Plates for the Reduction and Fixation of Mandibular Fractures

**DOI:** 10.3390/jcm10245975

**Published:** 2021-12-20

**Authors:** Marc Anton Fuessinger, Mathieu Gass, Caroline Woelm, Carl-Peter Cornelius, Ruediger M. Zimmerer, Philipp Poxleitner, Stefan Schlager, Marc Christian Metzger

**Affiliations:** 1Department of Oral and Maxillofacial Surgery, University Freiburg Hugstetterstr, 55, 79106 Freiburg, Germany; mathieu.gass@uniklinik-freiburg.de (M.G.); caroline.woelm@t-online.de (C.W.); philipp.poxleitner@uniklinik-freiburg.de (P.P.); stefan.schlager@anthropologie.uni-freiburg.de (S.S.); marc.metzger@uniklinik-freiburg.de (M.C.M.); 2Department of Oral and Maxillofacial Surgery, University Munich Lindwurmstr, 2a, 80137 Munich, Germany; peter.cornelius@med.uni-muenchen.de; 3Department of Oral and Maxillofacial Surgery, School Hanover Carl-Neuberg-Str. 1, 30625 Hannover, Germany; zimmerer.ruediger@med.uni-leipzig.de

**Keywords:** mandibular fractures, preformed osteosynthesis plates, virtual analysis

## Abstract

Purpose: The known preformed osteosynthesis plates for the midface are helpful tools for a precise and fast fixation of repositioned fractures. The purpose of the current study is to analyze the precision of newly developed prototypes of preformed osteosynthesis plates for the mandible. Methods: Four newly designed preformed osteosynthesis plates, generated by a statistical shape model based on 115 CT scans, were virtually analyzed. The used plates were designed for symphyseal, parasymphyseal, angle, and condyle fractures. Each type of plate has three different sizes. For analysis, the shortest distance between the plate and the bone surface was measured, and the sum of the plate-to-bone distances over the whole surface was calculated. Results: A distance between plate and bone of less than 1.5 mm was defined as sufficient fitting. The plate for symphyseal fractures showed good fitting in 90% of the cases for size M, and in 84% for size L. For parasymphyseal fractures, size S fits in 80%, size M in 68%, and size L in 65% of the cases. Angle fractures with their specific plate show good fitting for size S in 53%, size M in 60%, and size L in 47%. The preformed plate for the condyle part fits for size S in 75%, for size M in 85%, and for size L in 74% of the cases. Conclusion: The newly developed mandible plates show sufficient clinical fitting to ensure adequate fracture reduction and fixation.

## 1. Introduction

Maxillofacial injuries are found to be among the major health problems worldwide, with their epidemiology varying widely in different countries [1,2,3]. Maxillofacial fractures occur in a significant proportion of trauma patients as a result of the prominent and exposed position of the head, and they have been increasing over the past decades [4]. The fractures are usually associated with dysfunctions in communication (speech and facial expressions), nutrition, breathing, hearing and vision, and with cosmetic consequences, causing emotional distress [5]. In the literature, the incidence of mandibular fractures is found to be from 65% to 70.5% of cases, strongly correlating with younger men between 20 and 30 years of age. The main causes of mandibular fractures are traffic accidents (car, bike, or motorcycle), violence, and falls [6]. These mechanisms combined cause over 80% of the fractures [4,7]. As for the location of the fractures, the literature shows mixed data. The four most common areas are, in that order, the body, the symphysis/parasymphysis, the condyle, and the angle [4,7]. The therapy of a fracture of the mandible can be a conservative or an operative one, depending on factors such as the type of the fracture, the anatomic location, and the patient’s dental status and age [8,9,10,11,12]. The goal of a surgical approach with osteosynthesis of the fracture is to obtain an exact repositioning of the fracture and the occlusion, as well as an early regaining of functionality [13]. Today, plate-and-screw osteosynthesis by means of manually bent plates for mandibular fractures is a standard procedure in routine clinical practice [14,15,16]. The development in the field of osteosynthesis plates shows two areas. One area is concerned with patient-specific implants, the other with preformed plates that achieve a satisfactory fit based on statistical methods. Both procedures are used in clinical routine. Both procedures have their advantages and disadvantages. Preformed osteosynthesis plates for the mandible and the orbital cavity have shown advantages in terms of fitting accuracy and less time consumption, allowing less invasive approaches with, as a result, fewer surgical risks as, e.g., nerve injury or postoperative hemorrhage [17,18,19,20,21]. Patient-specific implants are highly precise and allow the patient’s individual anatomy to be taken into account. The dimensions of the implants can thus be designed to be as stable as necessary and as delicate as possible.

The accuracy of fit of the preformed plates was defined as the main measurement parameter.

A high degree of accuracy of fit is required for clinical use. Manual adaptation should be avoided for the most part, on the one hand, to generate real-time savings and, on the other hand, to prevent perforation through the soft tissues and dislocation of the fracture fragments. Small inaccuracies in the fit can be compensated to a certain extent with the help of angular stable plates and their screws. With the help of this screw technique, the fracture pieces are not pulled to the plate, and the repositioning of the fracture is not disturbed. In addition to the known reconstruction plate of the mandible (MatrixMANDIBLE preformed reconstruction plate), the purpose of this study is to determine the fitting accuracy of preformed osteosynthesis plates for non-comminuted fractures of the mandible by virtual testing.

## 2. Materials and Methods

Virtual testing of four different types of preformed plates in several different sizes (condyle plate, sizes S, M, L; body plate, sizes S, M, L; angle plate, sizes S, M, L; symphysis plate, sizes M, L) was performed on the left and right sides of 50 CT scans of Europeans (30) and Asians (20), using the 3D software GOM Inspect V8 (GOM GmbH, Braunschweig, Germany) [22]. CT scans were performed in cases of suspected midface fracture as a result of a fall, brute force trauma, or accident. In total, 1000 tests were performed (20 plates per patient). Exclusion criteria were pathological findings (fractures, deformities, excessive atrophy) and artifact-causing foreign bodies (dental implants, MFC crowns, screws) to avoid an additional source of error.

### 2.1. Design of the Plates

The shapes and sizes of the plates were designed according to the requirements of the AO Foundation. For simple linear fractures, the AO demands load-sharing osteosynthesis. A reduced buttress demands a more stable osteosynthesis. The most frequently applied technique in mandibular fractures is miniplate osteosynthesis combining two plates on the upper and lower border of the mandible. To compensate for the tensile and the compression strain, we in our department prefer the two-plate approach. The upper plate is located in the tension zone, and the lower plate is placed in the pressure zone [23,24,25]. Following this principle, the suggested plates have an upper and a lower part, stabilized by connectors, which enlarge the possible screw positions and increase the overall stability without increasing the thickness of the plates. The newly introduced plates are connected to each other to heighten the stability without increasing the thickness of the used plates, in order to avoid soft-tissue irritation. By the modified design with connected miniplates, the application range could be enlarged. The preformed plates fitted the strain occurring in the described regions during functioning. The basic elements of the plates are already in clinical use. The combination of two plates and an enlarged material cross-section increases the mechanical properties, for which reason a finite element analysis (FEA) was not required for product approval.

The indication for the preformed plates varies from the simple healthy fracture site to multi-fragmentary fracture sites. Only fracture sites with an extensive defect demand reconstruction plates.

To generate the design of the preformed plates, a statistical model was used based on 127 mandibles from clinical CT scans, not included in the test group, with the following distribution regarding ethnicity, sex, and age: Europeans—37 in total, 14 males, 23 females, average age 46.9 y, SD age 21 y; Chinese—90 in total, 48 males, 42 females, average age 43.5 y, SD age 16 y. While the age of the Chinese patients is roughly normally distributed, the European patients show an under-representation in the age interval between 40 and 50 years and an over-representation of patients older than 70 years. After automated segmentation by use of Brainlab iPlan CMF 3.0 (Brainlab, Feldkirchen, Munich, Germany), 3D triangular surface meshes were generated. To ensure (pseudo)homology throughout the sample, an elastic registration procedure (for further details on registration, see [26]) was employed. All statistical analyses were done with the statistical software R [27] and, specifically, the R packages Morpho, Rvcg [28], and RvtkStatismo [29]. The procedure to compute surfaces representing an optimal shape was identical for all regions: The different regions of interest were defined based on the sample mean and were successively extracted automatically from all registered meshes, exploiting the identical mesh topologies (i.e., corresponding vertex indices). The vertices belonging to this region were then rigidly aligned using a Procrustes registration [30], and a principal component analysis (PCA) was computed on these aligned data. The first PCA represents the major axes of variation. As the data were not standardized regarding size, the first PCA is associated with allometric effects and sexual dimorphism. In order to model these effects, we regressed the shape of this region onto the first PCA, calculating a surface model for the 20%, 50%, and 80% quantiles of the first PCA scores. In that way, the resulting surfaces do not only vary isotropically but also incorporate the shape change associated with size.

### 2.2. Analysis of the Accuracy of the Plate

Analysis of the accuracy was done on 50 additional skulls not employed to establish the statistical shape model. As to the ethnical and gender distribution, 30 skulls were from Europeans (18 males; 12 females) and 20 skulls were from Asians (6 males; 14 females). They were segmented using Voxim IVS Solutions (IVS Technology GmbH i.L., Chemnitz, Germany) based on the Hounsfield segmentation and then exported as STL files.

Initially, a best-fit method was chosen (Figure 1, Figure 2, Figure 3 and Figure 4). Three points were positioned on every plate, defined as starting points. The same three points were placed on the surface of the mandible as ending points to position the plate and measure the distances between the two surfaces.

In a second step, manual optimization was performed to avoid malpositioning such as overlapping of the foramen mentale or surpassing the lower border of the mandible.

In a third step, with the tool “local best-fit with tolerances“, the software corrected the positioning by avoiding the plate diving into the surface of the mandible and by attaining the lowest possible spacing. This procedure was repeated for all 1000 tests. Subsequently, the measurement points were placed on the plates, and the distances between the two surfaces were measured. The software computed, from every point of the polygon on the backside of the plates, the vertical distance to the surface of the mandible (GOM Inspect V8 software) [31]. The measurement result of each testing was automatically saved in an Excel sheet. The statistical evaluation was performed using STATA 13 [32].

## 3. Results

Sufficient fitting of a preformed osteosynthesis plate is defined by the authors as a maximum distance of 1.5 mm from the bone surface to the plate surface [33]. More than 1.5 mm distance between plate and bone leads to soft tissue irritation, especially in the field of cranio-maxillo-facial surgery.

Depending on the plate size, the osteosynthesis plate of the condyle fits in 75% of the cases for size L, in 85% of the cases for size M, and in 75% of the cases for size S. The osteosynthesis plate of the mandible body shows a sufficient fit in 65% of the cases in size L, in 68% of the cases in size M, and in 80% of the cases in size S. In 47% of the cases in size L, in 60% of the cases in size M, and in 53% of the cases in size S, the osteosynthesis plate of the angle shows a sufficient fit. The osteosynthesis plate of the symphysis shows fitting values between 84% in size L and 90% in size M, with no outliers larger than 3 mm. There was no statistically significant difference regarding the different sides of application on the mandible (Table 1).

Ethnic differences only existed regarding the symphysis plate. In the Asian population, the plates fit in 100% of the cases, whereas in the European population, the plates fit in 78% of the cases.

The body plate fits for the male population in 81% of the cases versus 63% for the female population. Regarding the condyle and the angle plate, no differences could be detected.

In general, the symphysis plate fits best, followed by the condyle plate and the body plate. The angle plate shows worse fitting values compared to the other plate types.

### Application of the Plates

Two clinical cases are included in this article in order to visualize the application of a symphysis and a condyle plate. Figure 5 shows an OPT of a patient with the symphysis plate in place. The fracture of the mandible is approached via an incision in the vestibular fold of the lower jaw. After visual evaluation of the fracture, the fracture is repositioned and the plate is fixated. Because of a lingual gap, a second standard miniplate was applied in addition. Figure 6 shows an OPT of a patient with a combined fracture of the condyle and the parasymphysis.

## 4. Discussion

Preformed osteosynthesis plates are helpful tools in cranio-maxillofacial surgery. For simple fractures, preformed plates for the orbital cavity and the mandible reduce the time-consuming step of plate bending without worsening the fracture repositioning. Although focused on the orbital cavity, preformed plates provide a higher accuracy of the reconstructed contour and an easier use, and, by reducing the operation time, they reduce the cost [21]. The mandible shows a more complex anatomy. The differences between the sexes as well as the dependence on size are factors for the preformed plates that make standardization and, thus, preforming difficult [34]. The present study was performed virtually as previously described by Chrcanovic et al. [10] in a review in order to obtain a better statement on the accuracy of fit due to the large number of computational options.

In our opinion, the results of the virtual test as shown can also be applied to the plates for the lower jaw, and this was done in previous studies [25,35,36,37,38]. In the area of the symphysis, the strong contouring of the plate helps to avoid lingual gaping of the fracture gap [38]. In the area of the condylar process, the preformed plate supports fracture reduction despite the limited overview. In comminuted fractures, the preformed plate helps to reposition the fracture because of its predetermined shape. Studies using preformed plates in comminuted fractures show advantageous results [25,39,40]. Based on a newly developed statistical shape model, preformed panels can be developed for different ethnicities, different age groups, and different genders.

Considering the data basis, a sufficiently large variance in age, gender, and ethnicity was obtained in the present study by using the statistical form model. The number of CT scans used was sufficient to obtain a statistically significant result for interpreting the fitting of the plates. A critical point of our analysis is the first setting of the measurement points. The initial positioning is defined by the manufacturer after consultation of experienced surgeons in the developer team. The main positioning of the plates is then done automatically by the freeware, simply by identifying the shortest distance between the plate and the bone surface. Manual repositioning was only necessary in cases when the automated positioning algorithm had put the plates next to sensitive anatomical structures. The possible bias of virtual analysis based on manual positioning is reduced due to this automated algorithm, while former studies used only the manual positioning approach [41].

The design of the plates is critical for their clinical application. The design is based on observations on clinically used osteosynthesis plates for mandibular fractures. In the condyle, body, and symphysis area, the use of two plates is generally accepted [25,42]. In non-displaced simple fractures of the mandibular angle, the miniplate fixation on the oblique ridge according to Champy is the standard method. The Champy technique is a minimally invasive and fast method. In this case, a preformed plate would not be advantageous [43].

In comminuted or more displaced fractures, application of two plates supports the surgeon in improving the bone buttressing and alignment. For this reason, the design of the angle plate employs two connected plates according to the osteosynthesis principles of the AO Foundation. Under load-bearing conditions in complex fracture patterns, preformed reconstruction plates are already in clinical use and show advantages concerning the repositioning of fractures, a minimized risk of fatigue fractures, and reduced operation times [19,39].

In our study, in about 72% of the cases, the preformed plates could be applied in the clinic. Based on individual surgical experience and former studies, the optimal distance between bone and osteosyntheses plate should be 2 mm or less [33,44,45]. In the shown study, the authors defined a maximum distance between the bone surface and the plate surface of 1.5 mm. The distance limit of 1.5 mm from the bone to the plate surface is clinically relevant. Gaps larger than 1.5 mm will lead to soft-tissue irritation and the danger of wound infection by soft-tissue perforation. At the defined limit of 1.5 mm, the clinical precision for fracture repositioning is sufficient, especially when using locking screws to hold the bone in the correct position without putting pressure on the fracture ends.

The availability of the preformed plates is a big advantage over patient-specific implants (PSI), which constitute the most precise but also the most expensive method in bone reconstruction. In addition, preformed plates can also be used in emergency cases. In only 21% of the cases, small adjustments have to be done, which means reduced bending and, consequently, a lower risk of fatigue fractures. In contrast to off-the-shelf plates, the amount of bending for the worst-fitting plates is still reduced. In 69.4% of cases, the fit is perfect. The shown gender discrepancy in the results for the angle plate is due to the developmentally stronger expression of the masseteric tuberosity.

Regarding the results, size L shows the worst fitting of all plates. Sizes M and S show better fitting; thus, size L plates need some adjustments before these prototypes can be used regularly. Improving the fitting for larger plates requires more information during the construction process. The larger the plate, the more prominent the contours of the mandible will be. One strategy could be to change the position of the plate to avoid the direct localization on the strong curvatures such as the tuberositas. Another possibility could be to integrate more data of larger mandibles before the construction process. Later, the selection of the respective plate is clinically comparable to that of the preformed orbital plates, based primarily on the gender and secondarily on the body size. Women mostly receive a plate of size S, men mostly a plate of size M.

## 5. Conclusions

The presented study shows the possibility to use preformed osteosynthesis plates in a precision sufficient for clinical use.

Apart from the clinically sufficient precision, it is conceivable that region- and gender-specific preformed plates will be designed by the statistical model. An increase in precision is thus easily feasible.

The cost side is also attractive due to the use of an open source code to calculate the statistical shape model [46]. In addition to manufacturing costs, the importance of the intraoperative efficacy of preformed plates becomes clear when considering the number of mandibular fractures per year. A reduction in operating time through the use of preformed plates is clinically and economically relevant for the patient and the hospital in general [19,20,21,39,47].

3D printing will challenge the preformed plates in the future. Still, the virtual reconstruction of a mandibular fracture by a manual approach is technically demanding and time consuming [48]. The preformed plates will be a kind of bridging technology until virtual fracture reconstruction is automated and in-house 3D printing is sufficiently fast. Currently, the cost–benefit ratio is much better than fully individualized plates, and the time in which the plates can be used is unrivaled.

## Figures and Tables

**Figure 1 jcm-10-05975-f001:**
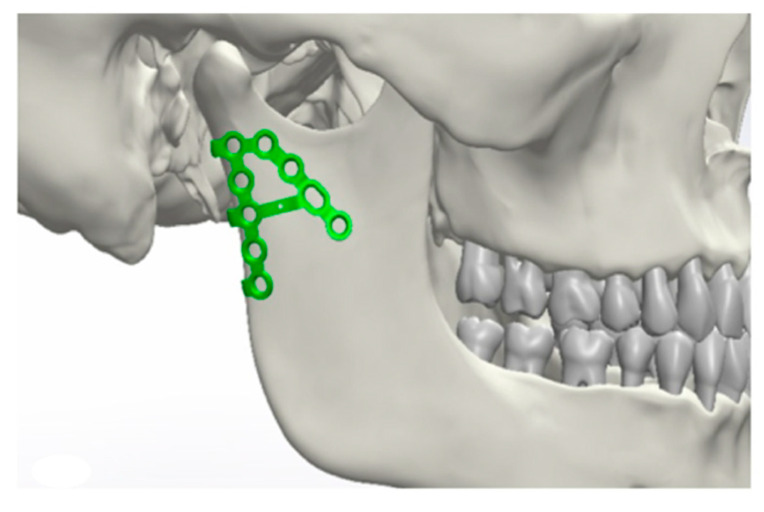
Positioning of the osteosynthesis plates in the area of the condyle.

**Figure 2 jcm-10-05975-f002:**
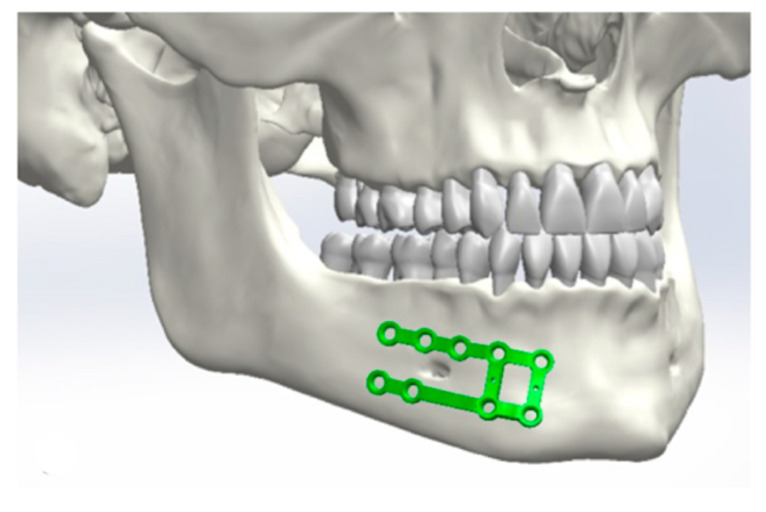
Positioning of the osteosynthesis plates in the area of the body.

**Figure 3 jcm-10-05975-f003:**
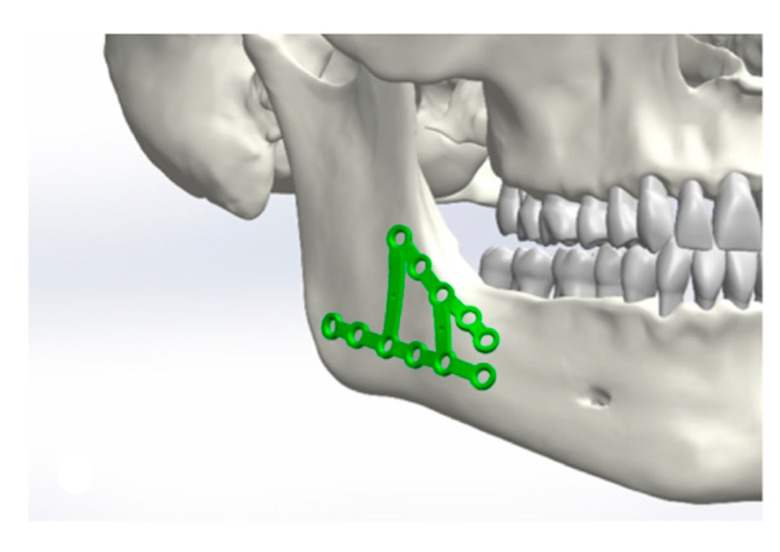
Positioning of the osteosynthesis plates in the area of the angle.

**Figure 4 jcm-10-05975-f004:**
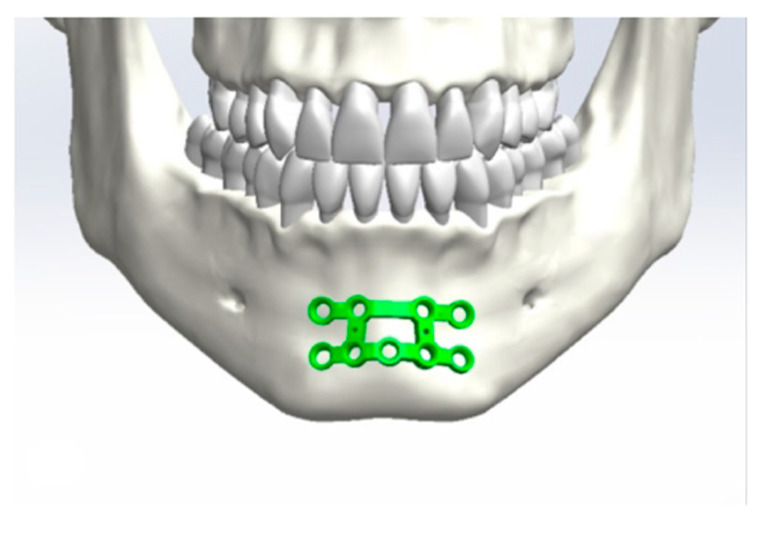
Positioning of the osteosynthesis plates in the area of the symphysis.

**Figure 5 jcm-10-05975-f005:**
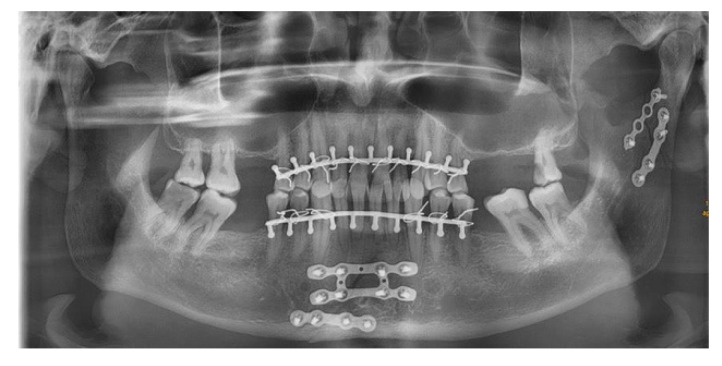
Visualization of the application of the symphysis plate.

**Figure 6 jcm-10-05975-f006:**
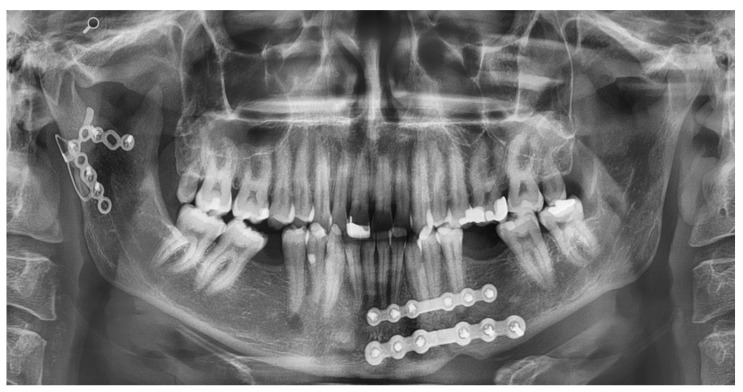
Visualization of the application of the condyle plate, with a modification to hold the upper segment.

**Table 1 jcm-10-05975-t001:** Fitting accuracy in percentage of the preformed osteosynthesis plates of the mandible.

	Size L	Size M	Size S	Total
Condyle plate	74	85	75	78
Body plate	65	68	80	71
Ramus plate	47	60	53	53
Symphysis plate	84	90	-	87

## Data Availability

The data presented in this study are available on request from the corresponding author. The data are not publicly available due to intellectual property.

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
