# Peer review of "Analyzing the Fitting of Novel Preformed Osteosynthesis Plates for the Reduction and Fixation of Mandibular Fractures"

_jcm, 2021, doi:10.3390/jcm10245975_

Round 1

Reviewer 1 Report

In this article, the authors' aim was to describe the novelty of their newly developed prototypes of preformed osteosynthesis plates. However, to my knowledge, 3-dimensional miniplates are not new and many different designs have been described in the literature.

Introduction is poorly written and does not exploit the subject properly, there should be more information about possible treatments and their advantages and disadvantages, new developments and designs, especially custom- made fixation plates manufactured with the use of different technologies.

Do the preformed plates resolve other problems than fitting? What kind of plates were used- were they compressive plates or other? Why the perfect fitting is needed (or not) – these should be described in the paper.

The exclusion criteria are wrong and should be clearly described- how the bone fracture may be included in the exclusion criteria in research about bone fracture plates  especially when the authors name the indications  in lines 83-85? If the CTs had been obtained from healthy patients, what was the reason to perform CT scans? The whole section should be rewritten.

Was the testing performed on one CT scan (50) and the mandible model on other CT scans (127)?

The design of the plate should be more extensively described- instead the authors have focused too much on technical details of  the computer software.

“Sufficient fitting of a preformed osteosynthesis plate is defined by the authors as a maximum distance of 1.5 mm from the bone surface to the plate surface”. – this needs explanation and be supported by relevant references. What dose it mean that the plate fits in 50% or 70 %? How does it impact the clinical outcome?

Also the literature is poorly described with no up to date literature and a lot of biased statements without scientific support. Basically this paper resembles more a technical report than a scientific research. There is practically no statistics or anything to compare.  In my opinion, this article in its present form has no clinical or scientific importance. With so many authors we should expect much more.

If the authors are able to improve it, maybe more technical journal like Applied Sciences will be more suitable for it.

Author Response

Dear Reviewer, 

thank you very much for the very helpful feedback. Every feedback helps us to improve the manuscript. 

Following, we answered your remarks:

In this article, the authors' aim was to describe the novelty of their newly developed prototypes of preformed osteosynthesis plates. However, to my knowledge, 3-dimensional miniplates are not new and many different designs have been described in the literature.

Introduction is poorly written and does not exploit the subject properly, there should be more information about possible treatments and their advantages and disadvantages, new developments and designs, especially custom- made fixation plates manufactured with the use of different technologies.  We add some more information in Line 54-57 and 61-63.

Do the preformed plates resolve other problems than fitting? In line 58-60, we described the further advantages of the plates.

 What kind of plates were used- were they compressive plates or other? Why the perfect fitting is needed (or not) – these should be described in the paper.  We add some more information about angular screws and the topic of plate fitting in Line 61-69.

The exclusion criteria are wrong and should be clearly described- how the bone fracture may be included in the exclusion criteria in research about bone fracture plates  especially when the authors name the indications  in lines 83-85?--> to avoid a bias by fractured mandibles they are not inlcuded, because the topic was just fitting accuracy of the plates.  

If the CTs had been obtained from healthy patients, what was the reason to perform CT scans? The whole section should be rewritten.

Was the testing performed on one CT scan (50) and the mandible model on other CT scans (127)? There are different CT Scans. The one group fort he desing process, and the other group fort he testing group (117)

The design of the plate should be more extensively described- instead the authors have focused too much on technical details of  the computer software.-> We add information for the design process (90-94

“Sufficient fitting of a preformed osteosynthesis plate is defined by the authors as a maximum distance of 1.5 mm from the bone surface to the plate surface”. – this needs explanation and be supported by relevant references. What dose it mean that the plate fits in 50% or 70 %? How does it impact the clinical outcome? We add the reasons for setting the limit on 1.5mm, based on literature and our experience, especially in the midface, there is less soft tissue coverage, because we reduce the recommend distance of <2mm to <1.5mm.

Also the literature is poorly described with no up to date literature and a lot of biased statements without scientific support. Basically this paper resembles more a technical report than a scientific research. There is practically no statistics or anything to compare.  In my opinion, this article in its present form has no clinical or scientific importance. With so many authors we should expect much more.--> we add literature concerning the distance between plate and bone (Ahmad et al.) and Literature for virtual testing of fitting accuracy (Tkanya et al., Schmutz et al) and the need for stable 3d-plates for fractures of the mandible. (de olveira et al.)

If the authors are able to improve it, maybe more technical journal like Applied Sciences will be more suitable for it.

We hope, we could meet your expectations.

Best regards 

Füßinger

Reviewer 2 Report

The study evaluates the precision of newly developed prototypes of preformed osteosynthesis plates to apply in cases of maxillofacial fractures. The authors designed by a statistical shape model based on 115 CT scans four types of preformed osteosynthesis plates, each for a different type of fracture: symphyseal, parasymphyseal, angle and condyle fractures. Each type of plate was developed for three different sizes (S,M,L). The quality of the implant fitting was evaluated by measuring the shortest distance between the plate and the bone over the whole surface and a distance less than 1.5 mm was considered to be sufficient fitting. Success rates differed between distinct types and sizes of plates, ranging between 47% and 90%. In all the plate types, size L showed the worst fitting so that the authors themselves considered that additional adjustments must be done before these prototypes can be used regularly. Nevertheless, the authors concluded that the newly developed mandible plates have sufficient clinical fitting to ensure adequate fracture reduction and fixation.

The paper is interesting from a medical point of view since and the manuscript contains sufficient noteworthy information to justify publication. The subject is significant because maxillofacial fractures occur in many trauma patients. The manuscript is concisely written and easy to understand. The statistical analysis been performed appropriately, the interpretation of the results are justified and the conclusions drawn are supported by the data and the adequate referencing of past studies. I have only two minor points to improve the manuscript:

  1. The table 1 could be better organized.
  2. The authors should further discuss about strategies to improve results in size L plates.
  3. Likewise, the authors should further discuss about the future applicability of the described method

Author Response

Dear Reviewer, thank´s a lot for the nice advices. I comment your ideas, direct to the three points: 

  1. The table 1 could be better organized.
    1. In Line 170-172 we cleared the table, the results are still the same.
  2. The authors should further discuss about strategies to improve results in size L plates.
    1. In Line 242-246, the problem with the plates in Size L were discussed and futre improvements are named.
  3. Likewise, the authors should further discuss about the future applicability of the described method
    1. In Line 240-243, a crosslink to trauma surgery was taken

Round 2

Reviewer 1 Report

Line 54- there is a spelling mistake: The development

Line 60- nerve bleeding?

Line 61- allowing you?- it needs paraphrasing

Lines 67- 70- these sentences are unclear and need rewriting; perforation of the osteosynthesis plate by the soft tissues?

line 71- angular stable screw or sliding screw or other?

The exclusion criteria should include all the features that may alter CT scans, e.g. dental implants, MFC crowns or others.

lines 92-93- reference missing

line 157- as above ref.

Table 1. needs to be corrected- are the numbers percentages? 

Additionally, it seems that Authors have confused discussion with conclusion- this needs rewriting. Moreover, the number of references is really not enough to support results. 

Author Response

Dear Reviewer,

Thank´s a lot for your comments to improve our article. 

We tried to fulfill your expectation.

Best regards

Fuessinger

Line 54- there is a spelling mistake: The development

>>corrected

Line 60- nerve bleeding?

>>substituted by postoperative hemorrhage

Line 61- allowing you?- it needs paraphrasing

>> Patient-specific implants are highly precise and allow the patient's individual anatomy to be taken into account. The dimensions of the implants can thus be designed to be as stable as necessary and as delicate as possible.

Lines 67- 70- these sentences are unclear and need rewriting; perforation of the osteosynthesis plate by the soft tissues?

>> The accuracy of fit of the preformed plates was defined as the main measurement parameter.

A high degree of accuracy of fit is required for clinical use. Manual adaptation should be avoided for the most part. On the one hand to generate a real time saving and on the other hand to prevent perforation through the soft tissues and dislocation of the fracture fragments..

line 71- angular stable screw or sliding screw or other?

>>Small inaccuracies in the fit can be compensated to a certain extent with the help of angular stable plates and their screws

The exclusion criteria should include all the features that may alter CT scans, e.g. dental implants, MFC crowns or others.

>> Exclusion criteria were pathological findings (fractures, deformities, excessive atrophy) and artifact-causing foreign bodies (dental implants, MFC crowns, screws) to avoid an additional source of error.

lines 92-93- reference missing

>>added referecnes from 2015-2021 (de Medeiros RC et al. 2015 &  2016 and Xu et al. 2021)

line 157- as above ref.

>>Ostas et al. 2021

Table 1. needs to be corrected- are the numbers percentages? 

>> Fitting accuracy in percentage of the preformed osteosynthesis plates of the mandible.

Additionally, it seems that Authors have confused discussion with conclusion- this needs rewriting. Moreover, the number of references is really not enough to support results. 

>> we rewrited the discusion part and add a conclusion. Moreover more references are added to support the results of the study

This manuscript is a resubmission of an earlier submission. The following is a list of the peer review reports and author responses from that submission.